# Regular Exercise and Weight-Control Behavior Are Protective Factors against Osteoporosis for General Population: A Propensity Score-Matched Analysis from Taiwan Biobank Participants

**DOI:** 10.3390/nu14030641

**Published:** 2022-02-02

**Authors:** Chih-Yi Hsu, Chun-Ying Huang, Ching-Hua Hsieh, Peng-Chen Chien, Chih-Chun Chen, Shao-Yun Hou, Shao-Chun Wu

**Affiliations:** 1Department of Anesthesiology, Kaohsiung Chang Gung Memorial and Chang Gung University College of Medicine, No. 123, Ta-Pei Rd., Niao-Song Dist., Kaohsiung City 833401, Taiwan; hsuchiyi@cgmh.org.tw (C.-Y.H.); b9702062@cgmh.org.tw (C.-C.C.); walkings@cgmh.org.tw (S.-Y.H.); 2Department of Trauma Surgery, Kaohsiung Chang Gung Memorial and Chang Gung University College of Medicine, No. 123, Ta-Pei Rd., Niao-Song Dist., Kaohsiung City 833401, Taiwan; junyinhaung@yahoo.com.tw; 3Department of Plastic Surgery, Kaohsiung Chang Gung Memorial and Chang Gung University College of Medicine, No. 123, Ta-Pei Rd., Niao-Song Dist., Kaohsiung City 833401, Taiwan; m93chinghua@gmail.com (C.-H.H.); VENU_CHIEN@hotmail.com (P.-C.C.)

**Keywords:** osteoporosis, osteoporotic fracture, exercise, weight control, Taiwan biobank, dual-energy X-ray absorptiometry

## Abstract

The rising prevalence of osteoporosis, which can lead to osteoporotic fractures, increases morbidity, mortality, and socioeconomic burden. Multiple factors influencing bone mass have already been identified. The aim of this study was to investigate whether exercise habits and weight-control behaviors can lower the incidence of osteoporosis in the general population. This retrospective study recruited all participants aged 35–70 years who underwent dual-energy X-ray absorptiometry (DXA) from Taiwan Biobank (TWB). The final analysis consisted of 3320 eligible participants divided into two groups; demographic characteristics, prevalence of clinical symptoms, comorbidities, and daily behavior were collected using a self-reported questionnaire. After propensity score matching with a 1:1 ratio, 1107 out of 2214 individuals were classified into the osteoporosis group. Age, body fat rate, body shape, diabetes mellitus, and social status were found to affect the incidence of osteoporosis. Subjects with a habit of regular exercise and weight-control behavior showed decreased odds of osteoporosis. (odds ratio: 0.709 and 0.753, 95% confidence interval: 0.599–0.839 and 0.636–0.890). In the general population, regular exercise or weight-control behavior lowers the incidence of osteoporosis.

## 1. Introduction

Osteoporosis is a major public health problem, and a projected rising prevalence is expected in an aged society [1]. However, it is often asymptomatic until osteoporotic fractures occur [2]. According to statistics from the International Osteoporosis Foundation (IOF), 158 million adults aged 50 years and above are at high risk of osteoporotic fracture worldwide, and the number is predicted to double by 2040. Osteoporosis is prevalent in women and the elderly, but it also occurs in all populations and at all ages, and osteoporotic fractures cause significant morbidity, mortality and psychosocial and financial consequences [3]. In the UK, the National Health Service is estimated to spend GPB 4.4 billion per year because of fragility fractures [4]. In the USA, the cost of treatment for osteoporotic fractures is expected to reach USD 95 billion annually by 2040 [5]. Each fracture case costs more than NTD 80,000 for initial treatment in Taiwan [6]. Mostly, surgical intervention is indicated, including open reduction internal fixation or bipolar hemiarthroplasty. After that, family support and rehabilitation are also needed. Therefore, early diagnosis and prevention are vital to healthcare systems [7].

Based on the definition from IOF, osteoporosis is diagnosed as a hip-bone mineral density (BMD) T-score of −2.5 or lower. Therefore, the value of bone mass is an objective target to prevent fracture because low bone mass is a strong risk factor [8]. For aged societies, such as Taiwan, increasing attention has been paid to osteoporosis over the years [9]. According to statistics from the 2005 to 2008 National Nutrition Survey in Taiwan, the prevalence of osteoporosis was 23.9% and 38.3% for men and women aged over 50 years, respectively [10]. Osteoporosis involves genetic, nutrient, socioeconomic status (SES), and behavioral factors. By the National Health and Nutrition Examination Survey, low SES is associated with risk factors for osteoporosis [11]. People with low SES are unaware of BMD status and are delayed for initiation of medical nutrient therapy after bone-density screening [12,13].The influence of behavioral factors, such as physical activity, smoking, and alcohol consumption, on bone health has been studied [14,15,16].

Finally, osteoporosis also elevates hip fracture risk. Besides falling, which is the major direct cause of hip fracture, hip fracture is also related to other risk factors, such as physical inactivity, low dietary calcium intake, diabetes, and Caucasian race [17,18,19,20]. Nevertheless, not all these factors are strongly associated with osteoporosis based on limited evidence. Previous studies did not find consistent association between diabetes and BMD [21,22,23]. There is also no absolute inverse correlation between hip fracture risk or incidence of osteoporosis and increase in dietary calcium intake, but very low calcium intake is indeed connected to a higher rate of fracture [24]. Though the Asian population has a lower BMD, lower incidence of hip fractures was found compared to Whites in the United States [25]. After a decade, hip geometry was proven as an independent factor contributing to hip fracture [26]. Among these factors, physical inactivity is consistently proven as a modifiable risk factor for either osteoporosis or bone fracture [27,28]. Exercise has already been proven and widely recommended for improving bone health [15]. Mechanical stimuli, such as muscle forces and ground reaction forces, increase the density and strength of bone minerals. Strength exercises prevent age-related or post-menopause-related bone loss [29,30,31]. Most importantly, intensity is emphasized. Walking is not as effective as resistance training alone or in combination with impact-loading activities in osteoporosis prevention [30,31]. Walking only provides a modest increase in the load on the skeleton above gravity. In addition to strength or aerobic exercise, a balance training program is also beneficial for women with established osteoporosis by reducing the incidence of falling [32].

However, previous studies have focused on specific groups, including older adults and postmenopausal women, who are at high risk for osteoporosis. Different from those studies, young adults (35–49 year old) were also recruited to our study. Among this age group, a very low percentage of people fulfill the criteria of osteoporosis [33]. These young patients usually suffer from chronic disorders or take long-term medication affecting bone metabolism, such as endocrine disorders, thalassemia, glucocorticoid, etc. Less commonly, genetic or idiopathic conditions can be found [34,35]. Patients’ conditions or comorbidities can directly or indirectly affect their exercise capability and intensity. Whether the exercise is resistance training or not, the aim of this study was to investigate if the general population with a habit of regular exercise has a lower incidence of osteoporosis after propensity score matching through the Taiwan Biobank (TWB).

## 2. Materials and Methods

### 2.1. Setting and Participants

The study was approved by the Institutional Review Board of Chang Gung Memorial Hospital (IRB No. 201800396B0). The IRB also approved the waiver of the participants’ consent. This research was based on the largest national biobank, Taiwan Biobank, collecting health data, questionnaires, and biological specimens, released through de-identified personal information. A total of 3320 individuals were recruited to our study cohort.

### 2.2. Study Design and Data Collection

The participants were included in the present study if they met the following criteria: (1) participants aged 35–70 years and (2) participants underwent dual-energy X-ray absorptiometry (DXA) for BMD measurement. Individuals with a history of cancer were excluded from the study. The questionnaire-based information included epidemiological characteristics, systemic disease, symptoms, lifestyle, social status, and nutritional factors (dietary and use of mineral or vitamin supplements). As part of the questionnaire, participants were also asked if they exercised regularly. Regular exercise was defined as a participant having an amount of exercise activity over 30 min each time, at least 3 times per week. A current alcohol drinker was defined as drinking 150 mL of alcohol per week for more than six months. Participants who smoked for more than six months were defined as voluntary smokers.

Participants were then divided into two groups for comparison according to their T-scores. The osteoporosis group was defined as a T-score ≤ −2.5. Participants with a T-score > −2.5 were put in the non-osteoporosis group. A total of 1660 participants with osteoporosis diagnosed by DXA were first retrieved from Taiwan Biobank. After that, another 1660 participants without osteoporosis were randomly selected from the database following the same sex ratio as osteoporosis group. Finally, 3320 individuals underwent analysis.

### 2.3. Taiwan Biobank (TWB)

The TWB is conducted by the Taiwanese government and focuses on providing researchers with collaboration opportunities [36]. Several cohorts are included for population diversity, and the database is updated gradually, including a large-scale community-based cohort and hospital-based cohorts. The community-based cohort study was planned to recruit 200,000 volunteers between 30 and 70 years of age with no history of cancer. Until July 2021, 148,567 volunteer participants were recruited from three collecting sites evenly allocated in northern, southern, and eastern Taiwan. In contrast, the hospital-based cohort study aimed to recruit 100,000 patients with the most common chronic diseases in Taiwan, including lung, breast, oral cavity and colorectal cancers, hepatitis, cardiovascular disease, diabetes, chronic kidney disease, stroke, Alzheimer’s disease, endometriosis, and asthma, from cooperating medical centers. Thus far, 7306 volunteer participants have been included in the project. Each subject in the TWB signed an approved informed consent form. Data collection was performed according to relevant guidelines and regulations. Apart from blood samples and physical examination, TWB researchers also conducted face-to-face interviews with participants and completed a structured questionnaire on personal information and lifestyle factors. According to official statistics, a total of 121,004 valid and available samples are in TWB currently. For demographic characteristics, the age group of 50–59 makes up the largest proportion of the age group distribution, accounting for 29.92% of the total. The male-to-female ratio is 1:1.8. In all, 57.94% subjects have studied in college, including those who already graduated. Overall, 60.49% female participants have habit of exercising, and 57.69% males do.

### 2.4. BMD Measurement

DXA scans of the central skeleton at the lumbar vertebrae were used to measure the BMD for the TWB as the gold-standard method by World Health Organization. In this case, the participants with metal lumbar implant, severe joint degenerative disease, and atherosclerosis of aorta were advised not to receive the examination. In addition to diagnosing osteoporosis, it has the advantage of assessing patients’ risk of fracture and monitoring response to treatment. Although some participants’ BMD was also measured using quantitative ultrasound (QUS) devices because of easy administration and lower price, this type of measurement correlated poorly with central DXA [37].

### 2.5. Statistical Analysis

All statistical analyses were performed using SPSS version 22.0 software (IBM Corp., Armonk, NY, USA), and the significance level for the two-tailed tests was set at 0.05. Data were tabulated as mean and standard deviation (SD) for quantitative variables and as absolute numbers. The Kolmogorov–Smirnov test was used to analyze the distribution of the variables. Categorical variables are presented as numbers. Comparisons between categorical groups were made using the chi-square analysis. The variables of demographic and clinical characteristics, SES, and lifestyles, which were significantly different between osteoporosis group (*n* = 1660) and non-osteoporosis group (*n* = 1660), were used in the binary logistic regression model to generate the propensity score. Using the propensity score, each group of patients was matched in a 1:1 ratio using Greedy’s nearest neighbor method (caliper of 0.2). Logistic regression analysis was performed to estimate the odds ratios (ORs) and 95% confidence intervals (CIs) of osteoporosis in relation to clinical characteristics, such as regular exercise and weight-control behavior.

## 3. Results

Of the 3320 subjects who received DXA for BMD measurement, 1660 people were diagnosed with osteoporosis according to a T-score ≤ −2.5. The remaining 1660 people were referred to the non-osteoporosis group. To minimize the differences in these two groups’ baseline demographic and clinical characteristics and lifestyle, propensity score matching was used. The variables that we included in propensity score matching were age, weight, fat body rate, waistline, hipline, diabetes, vertigo, cataract, heart rate, red blood cell (RBC) count, hemoglobin (Hb), hematocrit (Hct), fasting blood sugar, and uric acid. For SES and lifestyle, we also included education level, income, marital status, residence, and smoking in our propensity score-matching procedure. In total, 2214 people were finally analyzed after matching. Table 1 and Figure 1 show a comparison of demographic data and comorbidities between the two groups. The average age of the participants in the osteoporosis group was significantly higher. Participants whose T-score > −2.5 had higher body weight, fat body rate, waistline, and hipline. No significant differences were observed for most comorbidities. The prevalence of diabetes mellitus (DM) in the osteoporosis group was 7.5%, which was significantly lower than that in the non-osteoporosis group. Clinical symptoms, including vertigo, joint stiffness, neck pain, sciatic pain, and headache, were present in 6.1%, 28.1%, 31.9%, 9.8%, and 20.5% of the cases, respectively. Although the prevalence of musculoskeletal problems was high, no significant difference between the two groups was observed.

Heart rate, blood pressure, and blood tests, including complete blood count and clinical biochemistry, were measured, and the results are shown in Table 2. The osteoporosis group had a faster heart rate, lower hemoglobin, and lower hematocrit (*p* = <0.001, <0.001, and <0.001, respectively). Regarding the lipid profile, people without osteoporosis had lower low-density lipoprotein levels although the data of both groups were within the acceptable normal range. The osteoporosis group also had a lower average level of uric acid in the plasma.

In Table 3, social status, including income, education, marital status, and place of residence, were also compared. Income and education levels showed significant differences. A higher percentage of individuals without osteoporosis had a college degree (41.3%) than those with osteoporosis (35.8%). The place of residence was related to the T-score. Participants with osteoporosis included a higher proportion of people living in northern Taiwan where urbanization is greater.

A comparison of lifestyle habits, such as eating habits, smoking, alcohol drinking, dietary patterns, and exercise, is shown in Table 4. Over 60% participants cooked at home, and the frequency of eating out did not affect the incidence of osteoporosis. Details of dietary supplements were not collected using the structure questionnaire. There was no significant difference in the pattern of dietary supplement use between the two groups. Less than half of the patients who exercised regularly had osteoporosis. Osteoporosis occurred significantly less in individuals with regular exercise and weight control than in those who did not. After propensity score matching, individuals exercising regularly or controlling body weight had a lower risk of osteoporosis (Table 5, odds ratio: 0.709 and 0.753, respectively).

## 4. Discussion

Any reason that leads to an imbalance between bone resorption and formation causes osteoporosis. Factors for osteoporosis can be classified as modifiable or non-modifiable. Reducing modifiable risk factors, such as inadequate nutritional absorption, lack of physical activity, weight loss, cigarette smoking, alcohol consumption, air pollution, or stress, has been proven to prevent osteoporosis [38]. In this study, propensity score matching was used for known non-modifiable factors and a few modifiable factors for osteoporosis. As a result, regular exercise and weight control are both independent factors in preventing osteoporosis.

Evidence from many RCTs and meta-analyses showed that exercise training, such as weight-bearing impact exercise and progressive resistance training, can promote the bone health of children and adolescents, pre- and postmenopausal women, and older men [39,40,41,42]. In this study, the exercises were not limited to resistance or weight-bearing training. On the contrary, “exercise” was defined as activities during which the participant breathed deeply or perspired in the questionnaire. An open-ended question was asked to determine the exercises done by the participants. Most people chose aerobic exercise as a routine exercise. However, the participants could write down a maximum of three exercises.

Therefore, it was difficult to further investigate which exercise was potentially effective in preventing osteoporosis. Although regular exercise and body weight control were shown to be protective factors for osteoporosis, the regularity and self-discipline were not defined or further evaluated. Participants who stood on the positive side for these two questions cared more about their health condition than did others. As shown in Appendix A, a variety of exercises were performed. Although most exercises were classified as weight-bearing exercises, other aerobic exercises, such as swimming, biking, and tai-chi, which specifically improve cardiovascular function, balance, and body strength, were also options for participants.

An association was found between body weight and bone mineral density though the relationship was still controversial. Jensen et al. and Villareal et al. pointed out that weight loss is associated with a decrease in BMD [43,44]. As obesity increases, bone mineralization increases, leading to a reduced risk of osteoporosis and related diseases [45]. A recent systematic review and meta-analysis in 2020 also found obesity as a protective factor to osteoporosis [46]. On the contrary, some studies also have indicated that low BMD is strongly associated with increased percentage body fat (PBF) in adults with obesity [47,48,49]. To date, the mechanism of this correlation remains unclear although several explanations have been proposed. The acceptable hypothesis is that bone mass increases to accommodate the greater mechanical load from a larger body mass. However, weight loss does not adversely affect BMD as long as the muscle is maintained with exercise training. In contrast, changes in BMD were correlated with changes in thigh muscle volume [50]. Muscle was more strongly related to BMD than fat or body weight. Maintenance of lean tissue and prevention of fat mass gain may be important for maintaining bone health. According to a longitudinal cohort study, a loss of lean tissue (0.9%) and gain in fat (9%) occurred concomitantly with a decrease in bone mass (1.6%) over 5 years in men aged 25 to 96 years [51].

Type 2 diabetes mellitus is significantly associated with osteoporosis risk. A study by Lin et al. from Taiwan concluded a positive association between the presence of type 2 DM and the incidence of osteoporosis [52]. Another small sample, cross-sectional study revealed no significant difference in bone density between diabetic and nondiabetic subjects [53]. Nevertheless, DM was more common among participants without osteoporosis, in conflict with previous evidence. This could be explained by the fact that the comorbidities of participants were self-reported. Prediabetes could sometimes be mistaken for diabetes because we did not observe differences in HbA1C levels between obese individuals. According to the mean value of HbA1C, over half of the participants are in the status of prediabetes. This might affect the statistical results. Moreover, although fasting sugar in the non-osteoporosis group was higher, clinical significance was not observed, and the test is less reliable compared to HbA1C test due to human error.

In postmenopausal women, adipocytes contribute to the production of estrogen, which inhibits bone resorption by osteoclasts [54]. Increases in adipose tissue with increasing BMI in postmenopausal women result in increased estrogen production, osteoclast suppression, and a resultant increase in bone mass. Previous study also verified our findings, as it reported that people with a higher body fat rate had a lower percentage of osteoporosis [14,54].

Smoking is known to cause osteoporosis and bone fracture. Cadmium contained in tobacco has been shown to play a critical role in the underlying mechanism. Smoking is therefore a major source of cadmium exposure in smokers. Wallin et al. indicated that even low cadmium exposure increases the risk of low BMD and fractures. They found that 10 pack-years of smoking could shorten the time to first hip fracture by 3% [55]. Although the amount of smoking was not analyzed in our study, people who smoked voluntarily had a higher percentage of osteoporosis.

A recent meta-analysis reported that supplementation with calcium and vitamin D significantly reduced the risk of fractures and hip fractures by 15% and 30%, respectively [56]. A prospective study in Lyon Hospital proved that combined treatment with vitamin D and calcium supplementation is beneficial for postmenopausal women [37]. However, maintaining the correct dietary calcium phosphate is also emphasized. High calcium intake can cause the calcium paradox [57]. Satisfying the vitamin D requirements is important to obtain the best response for BMD. Vitamin D supplementation, with or without calcium, can increase bone mineral density (BMD) and have a positive anti-fracture effect. Vitamins and minerals are the main supplements taken by people in Taiwan [58]. However, the highest consumption was for vitamin E (19.4%), followed by vitamin C (9.7%) in the vitamins category. The most popular item was calcium (20.9%) in the mineral category. In people who took vitamins and minerals, only 2.1% consumed vitamin D with calcium [31]. The low usage of vitamin D plus calcium may explain why people using dietary supplements had no lower osteoporosis incidence.

A strong connection was found between socioeconomic status (SES) and health [59]. People with lower education and lower income have a greater risk of fracture [60,61]. A study based on the Korean adult population revealed that osteoporosis was more prevalent among men with a lower household income and women living in rural areas [62]. In a cohort study by Varenna et al., a protective role was played by increases in formal education [23]. An increase in educational status was associated with a significantly reduced risk of osteoporosis. Nevertheless, the study was limited because the researchers focused on a low educational level. They did not further stratify samples with more than 9 years of formal education. The results of our study showed that people with more than a high school education comprise a large proportion of osteoporosis patients. This is because the nine-year education program from primary to junior high schools started to be implemented in Taiwan since 1968. However, the percentage of osteoporosis between the high school and college education subgroups was not significantly different.

A study on osteoporosis between urban and rural areas was also conducted by Kang et al. The prevalence of osteoporosis in rural residents was higher than that in urban area residents. This tendency was also observed in our results. The urbanization level is highest in northern Taiwan, which leads to a lower percentage of osteoporosis [63].

There are some limitations to this study. Although this is a large database conducted by the Taiwanese government, over 95% of Taiwan’s population of 23.4 million consists of Han Chinese. Second, data collection for this cohort was based on a structured and rough questionnaire. Personal information and lifestyle factors were mostly self-reported. Common comorbidities except cancers were included without consideration of the nature of diseases. The severity of diseases could potentially impact osteoporosis. Moreover, the interaction between drug and osteoporosis was not considered. In addition, some of the questions were too general, imprecise, or indefinite. More than one option of exercise could be chosen by the participant, and the intensity among these exercises was also different. Therefore, the frequency and the intensity discrepancies between different exercise cannot be controlled during analysis in this study to make our result more convincing. These limitations should be noticed in future study.

## 5. Conclusions

After propensity score matching, most covariates and known risk factors were balanced. For the general population in Taiwan, regular exercise or weight-control behavior is an independent factor for lowering the incidence of osteoporosis.

## Figures and Tables

**Figure 1 nutrients-14-00641-f001:**
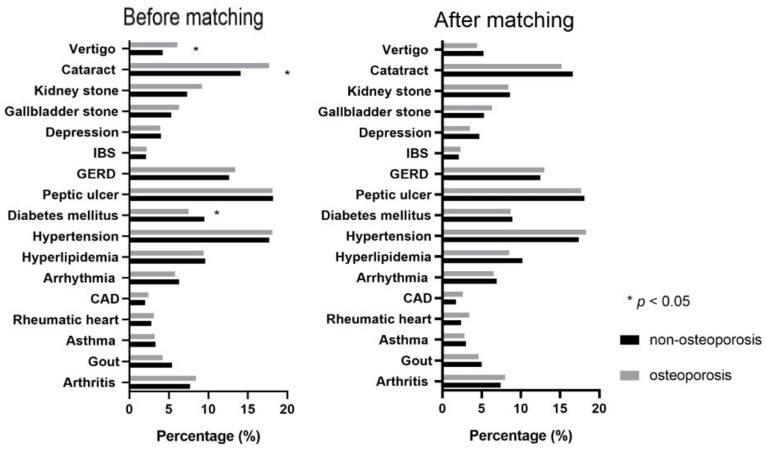
Comorbidities in osteoporosis and non-osteoporosis group.

**Table 1 nutrients-14-00641-t001:** Demographic characteristics of body compositions and medical history.

Variable	Before Matching			After Matching		
	T-Score > −2.5	T-Score ≤ −2.5	*p-*	T-Score > −2.5	T-Score ≤ −2.5	*p-*
	(*n* = 1660)	(*n* = 1660)	Value	(*n* = 1107)	(*n* = 1107)	Value
Age	57.08 ± 7.13	58.14 ± 7.03	<0.001	57.61 ± 6.87	57.79 ± 7.03	0.553
Male sex (%)	665(40.1)	665(40.1)	>0.999	435(39.3)	430(38.8)	0.862
Height (cm)	160.81 ± 8.06	160.28 ± 8.60	0.070	160.42 ± 7.91	160.56 ± 8.52	0.677
Weight (kg)	64.07 ± 11.01	61.14 ± 11.42	<0.001	62.47 ± 10.35	62.14 ± 11.51	0.478
Fat body rate (%)	28.91 ± 7.67	27.62 ± 7.26	<0.001	28.20 ± 7.24	28.17 ± 7.22	0.931
Waistline	85.64 ± 9.37	83.85 ± 9.41	<0.001	84.46 ± 8.77	84.59 ± 9.38	0.733
Hipline	96.65 ± 6.59	95.02 ± 6.33	<0.001	95.79 ± 6.21	95.75 ± 6.25	0.862
Allergic history (%)	165(9.9)	177(10.7)	0.530	109(9.8)	112(10.1)	0.887
Arthritis (%)	127(7.7)	139(8.4)	0.482	82(7.4)	89(8.0)	0.633
Gout (%)	89(5.4)	70(4.2)	0.143	55(5.0)	51(4.6)	0.765
Asthma (%)	55(3.3)	53(3.2)	0.922	33(3.0)	31(2.8)	0.899
Rheumatic heart (%)	47(2.8)	52(3.1)	0.683	27(2.4)	38(3.4)	0.208
CAD (%)	33(2.0)	40(2.4)	0.478	19(1.7)	29(2.6)	0.189
Arrhythmia (%)	104(6.3)	97(5.8)	0.662	76(6.9)	72(6.5)	0.799
Hyperlipidemia (%)	159(9.6)	156(9.4)	0.906	113(10.2)	94(8.5)	0.189
Hypertension (%)	293(17.7)	301(18.1)	0.751	193(17.4)	203(18.3)	0.618
Diabetes mellitus (%)	158(9.5)	125(7.5)	0.047	99(8.9)	96(8.7)	0.881
Peptic ulcer (%)	302(18.2)	301(18.1)	>0.999	200(18.1)	196(17.7)	0.868
GERD (%)	209(12.6)	223(13.4)	0.503	138(12.5)	144(13.0)	0.750
IBS (%)	35(2.1)	36(2.2)	>0.999	23(2.1)	25(2.3)	0.884
Depression disorder (%)	66(4.0)	64(3.9)	0.929	52(4.7)	39(3.5)	0.199
Gallbladder stone (%)	88(5.3)	105(6.3)	0.235	59(5.3)	70(6.3)	0.364
Kidney stone (%)	121(7.3)	152(9.2)	0.058	95(8.6)	93(8.4)	0.939
Vertigo (%)	70(4.2)	102(6.1)	0.015	58(5.2)	49(4.4)	0.428
Joint stiffness (%)	421(25.4)	467(28.1)	0.078	272(24.6)	304(27.5)	0.133
Neck pain (%)	501(30.2)	529(31.9)	0.311	342(30.9)	342(30.9)	>0.999
Sciatic pain (%)	164(9.9)	163(9.8)	>0.999	104(9.4)	112(10.1)	0.616
Headache (%)	304(18.3)	341(20.5)	0.114	206(18.6)	221(20.0)	0.451
Cataract (%)	234(14.1)	294(17.7)	0.005	184(16.6)	168(15.2)	0.383
Glaucoma (%)	41(2.5)	33(2.0)	0.411	23(2.1)	23(2.1)	>0.999
Dry-eye syndrome (%)	238(14.3)	263(15.8)	0.245	164(14.8)	171(15.4)	0.722
Retinal detachment (%)	33(2.0)	48(2.9)	0.115	26(2.3)	34(3.1)	0.360
Myodesopsia (%)	258(15.5)	277(16.7)	0.396	178(16.1)	181(16.4)	0.908

All values as mean SD or number and percent. CAD, coronary artery disease; GERD, gastroesophageal reflux disease.

**Table 2 nutrients-14-00641-t002:** Physiological parameters and biochemistry data.

Variable	Before Matching			After Matching		
	T-Score > −2.5	T-Score ≤ −2.5	*p-*	T-Score > −2.5	T-Score ≤ −2.5	*p-*
	(*n* = 1660)	(*n* = 1660)	Value	(*n* = 1107)	(*n* = 1107)	Value
SBP (mmHg)	122.01 ± 17.48	123.02 ± 18.78	0.110	122.77 ± 17.98	122.27 ± 18.97	0.532
DBP (mmHg)	73.15 ± 10.70	73.40 ± 10.96	0.500	73.21 ± 10.85	72.99 ± 10.91	0.629
Heart rate	34.90 ± 4.59	35.58 ± 4.83	<0.001	35.32 ± 4.73	35.09 ± 4.58	0.239
RBC	4.77 ± 0.51	4.68 ± 0.51	<0.001	4.73 ± 0.51	4.71 ± 0.49	0.592
WBC	5.75 ± 1.46	5.81 ± 1.57	0.226	5.77 ± 1.50	5.70 ± 1.44	0.249
Hb	14.00 ± 1.46	13.79 ± 1.47	<0.001	13.89 ± 1.48	13.87 ± 1.43	0.823
Hct	43.40 ± 4.33	42.78 ± 4.28	<0.001	43.03 ± 4.38	43.06 ± 4.17	0.855
Plt	229.18 ± 55.06	228.35 ± 54.89	0.663	229.63 ± 55.87	227.70 ± 54.20	0.409
HbA1c (%)	5.99 ± 0.96	5.96 ± 0.91	0.268	5.97 ± 0.89	5.98 ± 0.93	0.794
Fasting blood sugar (mg/dL)	101.10 ± 26.65	99.33 ± 24.38	0.047	100.54 ± 25.14	99.88 ± 24.97	0.536
Total cholesterol (mg/dL)	200.76 ± 36.95	200.11 ± 36.90	0.615	200.21 ± 36.99	200.54 ± 36.36	0.832
TG (mg/dL)	120.21 ± 86.33	123.99 ± 102.88	0.251	119.57 ± 75.25	118.61 ± 79.84	0.770
HDL (mg/dL)	54.14 ± 13.73	54.90 ± 13.99	0.119	54.62 ± 14.38	54.85 ± 13.47	0.693
LDL (mg/dL)	125.53 ± 32.10	123.23 ± 32.08	0.039	124.41 ± 32.49	124.57 ± 32.46	0.906
Total bilirubin (g/dL)	0.69 ± 0.29	0.68 ± 0.27	0.248	0.69 ± 0.30	0.69 ± 0.27	0.934
Albumin (g/dL)	4.54 ± 0.24	4.52 ± 0.23	0.008	4.53 ± 0.24	4.53 ± 0.23	0.510
GPT (U/L)	25.24 ± 20.33	24.13 ± 17.42	0.092	25.07 ± 20.26	24.06 ± 16.11	0.192
α-fetoprotein (U/L)	3.35 ± 3.83	3.48 ± 3.31	0.312	3.42 ± 4.55	3.37 ± 1.75	0.703
γ-GT (U/L)	25.63 ± 25.62	28.32 ± 52.69	0.062	26.19 ± 28.05	25.68 ± 27.43	0.662
BUN (mg/dL)	14.36 ± 3.93	14.17 ± 4.87	0.223	14.38 ± 3.88	14.37 ± 4.46	0.957
Creatinine (mg/dL)	0.75 ± 0.21	0.74 ± 0.39	0.169	0.75 ± 0.21	0.74 ± 0.36	0.333
Uric acid (mg/dL)	5.72 ± 1.37	5.47 ± 1.39	<0.001	5.60 ± 1.35	5.55 ± 1.38	0.345
Urine microalbumin (mg/dL)	32.86 ± 195.62	34.07 ± 142.98	0.838	28.14 ± 108.75	37.40 ± 163.67	0.117

All values as mean SD or number and percent. SBP, systolic blood pressure; DBP, diastolic blood pressure; RBC, red blood cell; WBC, white blood cell; Hb, hemoglobin; Hct, hematocrit; Plt, platelet, TG, triglycerides; HDL, high-density lipoprotein; LDL, low-density lipoprotein; GPT, glutamate pyruvate transaminase; γ-GT, γ-glutamyl transferase.

**Table 3 nutrients-14-00641-t003:** Socioeconomic, residency, and educational status.

Variable	Before Matching			After Matching		
	T-Score > −2.5	T-Score ≤ −2.5	*p-*	T-Score > −2.5	T-Score ≤ −2.5	*p-*
	(*n* = 1660)	(*n* = 1660)	Value	(*n* = 1107)	(*n* = 1107)	Value
Income (per month)	5.21 ± 3.44	4.97 ± 3.29	0.045	5.03 ± 3.41	5.10 ± 3.34	0.600
Education			<0.001			0.478
College	685(41.3)	595(35.8)		429(38.8)	440(39.7)	
Senior high school	541(32.6)	561(33.8)		367(33.2)	365(33.0)	
Junior high school	200(12.0)	228(13.7)		139(12.6)	127(11.5)	
Elementary school	218(13.1)	254(15.3)		158(14.3)	160(14.5)	
Illiteracy	16(1.0)	22(1.3)		14(1.3)	15(1.4)	
Marital status			<0.001			0.239
Married	1338(80.6)	1273(76.7)		870(78.6)	867(78.3)	
Single	64(3.9)	86(5.2)		51(4.6)	48(4.3)	
Divorced/separated/widowed	258(15.5)	301(18.1)		186(16.8)	192(17.3)	
Residence (%)			<0.001			0.855
Northern Taiwan	320(19.3)	392(23.6)		226(20.4)	248(22.4)	
Central Taiwan	401(24.2)	439(26.4)		278(25.1)	273(24.7)	
Southern Taiwan	911(54.9)	797(48.0)		584(52.8)	568(51.3)	
Eastern Taiwan	28(1.7)	32(1.9)		19(1.7)	18(1.6)	

All values as mean SD or number and percent. CAD, coronary artery disease; GERD, gastroesophageal reflux disease.

**Table 4 nutrients-14-00641-t004:** Dietary habits, healthy behavior, and healthcare seeking.

Variable	Before Matching			After Matching		
	T-Score > −2.5	T-Score ≤ −2.5	*p-*	T-Score > −2.5	T-Score ≤ −2.5	*p-*
	(*n* = 1660)	(*n* = 1660)	Value	(*n* = 1107)	(*n* = 1107)	Value
Cook at home	1034(62.3)	1001(60.3)	0.254	679(61.3)	685(61.9)	0.827
Eat late-night supper	378(22.8)	425(25.6)	0.062	252(22.8)	276(24.9)	0.251
Eat out			0.579			0.963
1 meal per day	290(17.5)	264(15.9)		183(16.5)	173(15.6)	
2–3 meal per day	197(11.9)	219(13.2)		132(11.9)	142(12.8)	
1–3 meals per week	424(25.5)	426(25.7)		269(24.3)	280(25.3)	
4–6 meals per week	118(7.1)	112(6.7)		78(7.0)	77(7.0)	
1–3 meals per month	499(30.1)	489(29.5)		347(31.3)	338(30.5)	
None	132(8.0)	150(9.0)		98(8.9)	97(8.8)	
Source of drinking-water			0.637			0.700
Well water	6(0.4)	11(0.7)		4(0.4)	9(0.8)	
Tap water	256(15.4)	271(16.3)		168(15.2)	171(15.4)	
Purified water	1191(71.7)	1161(69.9)		792(71.5)	789(71.3)	
Mineral water	75(4.5)	77(4.6)		55(5.0)	50(4.5)	
Others	132(8.0)	140(8.4)		88(7.9)	88(7.9)	
Drink tea	504(30.4)	510(30.7)	0.851	318(28.7)	346(31.3)	0.210
Drink coffee	545(32.8)	531(32.0)	0.630	361(32.6)	353(31.9)	0.750
Vegetarian			0.373			0.474
Used to be	65(3.9)	61(3.7)		44(4.0)	34(3.1)	
Yes	82(4.9)	100(6.0)		60(5.4)	65(5.9)	
No	1513(91.1)	1499(90.3)		1003(90.6)	1008(91.1)	
Drink alcohol			0.996			0.321
Quit	58(3.5)	59(3.6)		38(3.4)	35(3.2)	
Frequently	110(6.6)	110(6.6)		73(6.6)	57(5.1)	
No or rarely	1492(89.9)	1491(89.8)		996(90.0)	1015(91.7)	
Voluntary smoking	107(6.4)	196(11.8)	<0.001	92(8.3)	83(7.5)	0.529
Involuntary smoking	159(9.6)	179(10.8)	0.275	99(8.9)	103(9.3)	0.825
Chewing betel nut	22(1.3)	33(2.0)	0.174	18(1.6)	14(1.3)	0.593
Substance dependence	27(1.6)	31(1.9)	0.691	16(1.4)	19(1.7)	0.734
Taking dietary Supplement			0.989			0.638
Irregularly	367(22.1)	365(22.0)		247(22.3)	239(21.6)	
Regularly	601(36.2)	605(36.4)		406(36.7)	392(35.4)	
None	692(41.7)	690(41.6)		454(41.0)	476(43.0)	
Behavior of seeking Healthcare			0.342			0.456
Visit Chinese traditional Doctor	156(9.4)	161(9.7)		108(9.8)	106(9.6)	
Visit doctor	975(58.7)	957(57.7)		649(58.6)	652(58.9)	
Go to pharmacy	123(7.4)	116(7.0)		76(6.9)	66(6.0)	
Folk medicine	28(1.7)	27(1.6)		19(1.7)	18(1.6)	
Observation	226(13.6)	208(12.5)		158(14.3)	142(12.8)	
Others	226(13.6)	208(12.5)		158(14.3)	142(12.8)	
Regular exercise	963(58.0)	820(49.4)	<0.001	658(59.4)	564(50.9)	<0.001
Weight control	826(49.8)	670(40.4)	<0.001	547(49.4)	469(42.4)	0.001

All values as mean SD or number and percent. CAD, coronary artery disease; GERD, gastroesophageal reflux disease.

**Table 5 nutrients-14-00641-t005:** Individual risk of osteoporosis for T-Score ≤ −2.5.

	OR	95% CI of OR
Regular exercise	0.709	0.599–0.839
Weight control	0.753	0.636–0.890

OR, odds ratio; CI, confidence interval.

## Data Availability

The data presented in this study are available from the corresponding authors upon reasonable request.

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
