# Peer review of "Regular Exercise and Weight-Control Behavior Are Protective Factors against Osteoporosis for General Population: A Propensity Score-Matched Analysis from Taiwan Biobank Participants"

_nutrients, 2022, doi:10.3390/nu14030641_

Round 1

Reviewer 1 Report

Dear authors,

The authors' manuscript "Regular exercise and weight control behavior are protective factors against osteoporosis: A Propensity score-matched analysis from Taiwan biobank participants. " investigated the relationship between regular exercise or weight control behavior and the incidence of osteoporosis in the general population using a large number of cases and nearly homogeneous Asian data (more than 95% of Taiwan's 23.4 million population is Han Chinese). The results concluded that in the general population, regular exercise (not limited to resistance training or weight-bearing training) and weight management behaviors is an independent factor for lower the incidence of osteoporosis. It is also interesting to note the results that social status had an effect on the incidence of osteoporosis. However, the significance and position of this study is unclear in this manuscript because the motivation and hypothesis for this study based on previous studies are not appropriately stated. The manuscript needs to review related papers and describe them in the introduction concisely. In addition, there are unclear points in terms of the research and analysis methods of this study, and the manuscript needs to be revised regarding them.

 Hope it will help the authors in their future research.

Sincerely yours

Major concerns. 1)

The following statements in Line 54 in the introduction are important, and the organization and logical development of these statements are unclear.

Physical inactivity, a low dietary calcium intake, and falls have been long cited to be major risk factors for hip fractures in Asia.(Lau and Cooper 1996) It has been reported that Asians have a lower incidence of fractures than Caucasians.(Ho 1996; Nakamura, et al. 1994; Yan, et al. 2004) A systematic review of the metabolic syndrome and osteoporosis in Asian populations has also been published.(Sugimoto, et al. 2016) As the authors also cite, the relationship between regular exercise and bone health/ osteoporotic hip fractures has already been reported(Cheung and Giangregorio 2012; Lau, et al. 2001).

The authors state in line66 "Whether the exercise is resistance training or not, " and describe the discussion of these results can be found in line 185. Moreover, line 269 also mentions the differences of the intensity of the exercise as a limitation in this study. However, these descriptions seem inconsistent, and we do not understand the argument of this manuscript.

The authors also conclude that social status had an effect on the incidence of osteoporosis, but why did they include the content in the study? The motivation is not mentioned in the introduction. A discussion of the results is given under Line 247 in the Discussion, but it is unclear whether the discussion was derived from this study.

Cheung, A. M., and L. Giangregorio

               2012           Mechanical stimuli and bone health: what is the evidence? Curr Opin Rheumatol 24(5):561-6.

Ho, S. C.

               1996           Body measurements, bone mass, and fractures. Does the East differ from the West? Clin Orthop Relat Res (323):75-80.

Lau, E. M., and C. Cooper

               1996           The epidemiology of osteoporosis. The oriental perspective in a world context. Clin Orthop Relat Res (323):65-74.

Lau, E. M., et al.

               2001           The incidence of hip fracture in four Asian countries: the Asian Osteoporosis Study (AOS). Osteoporos Int 12(3):239-43.

Nakamura, T., et al.

               1994           Do variations in hip geometry explain differences in hip fracture risk between Japanese and white Americans? J Bone Miner Res 9(7):1071-6.

Sugimoto, T., et al.

               2016           Lifestyle-Related Metabolic Disorders, Osteoporosis, and Fracture  Risk  in  Asia: A Systematic Review. Value Health Reg Issues 9:49-56.

Yan, L., et al.

               2004           Does hip strength analysis explain the lower incidence of hip fracture in the People's Republic of China? Bone 34(3):584-8.

  2)

Regarding the data analysis described below the line 127, it is unnatural that exactly half of the population aged 35-70 years is prevalent when osteoporosis is defined as T-score -2.5 or less. Line 129 description "after propencity....." is not grammatically valid in English. Also, the description of the process of narrowing down the number of respondents from 3320 to 2214 in Line 131 is unclear. This part is important and needs to be described in detail.      

Minor comments. Line33-35

It is important to cite the appropriate references in the first 1-2 sentences in the introduction. It will be information that the authors probably consider to be well-known, but will let the beginners in this field and future readers know what kind of social context the study was conducted in.

Line 41-42

The authors focused on the medical economy in this section, and it is good that they compare the United States with Taiwan. However, many people may not understand the significance of the crude data of US$95 Billion in the US and US$3000 in Taiwan. It is necessary to correct the data using population, national budget, size, etc. to show the significance and importance of the numbers.

  M&Ms

The line position of the subheadings in M&Ms is inappropriate and difficult to read. (Because there is no line space between the previous paragraph and the paragraph below the subheading, and also because there is line space between the paragraph below the subheading and the paragraph before it)

  Line94

A detailed description of the proportion of community-based and hospital-based cohorts at the time of the survey and their demographic data is required.

Line108 2.4 BMD measurement

In Line 111, the authors state that "QUS devices because of easy administration and lower price, this type of measurement correlated poorly with central DXA”. Did the authors exclude QUS subjects? If not, how many of the subjects measured with them were included?

This is not mentioned in the limitation, but it may affect the reliability of the results.

The detailed information of the patient's demographic and clinical characteristics and lifestyles described on line 119 is required.

Reviewer 2 Report

Thank you for the opportunity to review this manuscript. The authors have presented a retrospective analysis of over 2200 individuals, chosen from a pool of 3320 eligible participants whose average age would fall out of the range of those more likely to be classified as osteoporotic. The indidivuals in the study are all exclusively Taiwanese citizens. The purpose of the study was to shift the focus of exercise and weight control's effects as protective against osteoporosis versus traditional studies that examine populations of elderly individuals who are most likely to be diagnosed as osteoporotic due to age-related predispositions.  The manuscript is written in very understandable English and it seems as though the analysis has included many subjects along with a large amount of DXA and questionairre data to process and describe.

Title: I do not feel this adequately describes the scope of the study.

Introduction
line 42-43 What is meant by the initial treatment in Taiwan.
Since the study is assessing individuals in the 35-70 year old range, then it might be important to establish, broadly, contributors to bone loss in the younger cohort instead of just the older post-menopausal group.

The introduction could be improved with additional information on obesity and other factors that could be contributory to bone loss (or not) in this age group before the discussion.

Methods and Results: The study has some major flaws, at least by my understanding of the methods and have several concerns with the idea of this study design. Firstly, the range of participants is nearly 40y and the retrospective analysis is not really retrospective since active DXA scans were requested. The exclusion of cancer patients is critical to this analysis, so that is a seemingly wise decision on the part of the researchers, however the inclusion of any other disease assumes they impose a significantly lower degree of bone loss which cannot be assumed. So, how does one decide to exclude one disease and include many others?

My other major concerns are the definition of exercise as only 30 min once over the course of a month. Additionally, the definition of 150mL of alcohol consumption per week for 6 months constituting alcohol consumption. Im not sure what that proves, because in contrast to non-drinkers, it is something quantifiable over time, but in contrast to alcoholics who drink in excess on a daily basis, this hardly constitutes a concerning risk factor for bone loss. 

The propensity scoring is never really discussed. This would help the reader understand what analysis was performed for grouping the subjects. 

Although it is mentioned in the abstract, it is not mentioned how many participants were grouped into the OP and non-OP groups after their T-score assessment in the methods section, which is where this must be described. To further complicate the understanding, the number of OP patients reported in the Results section is 1660. 

Were multiple DXA scans or questionnaires performed, as this might allude to the ability of exercise to maintain or improve bone, or reduce the impact of OP?

The authors should more accurately describe where "central body" scanning is for the DXA readings. 

Was any correlation run between obesity and BMD/T-scores?

Discussion
The statement in line 202 is very controversial. Increased body weight might correlate with increased bone, but many studies have indicated that increased weight as a consequence of obesity is not causal to increased bone mineral density, but detracts from bone quality. 

Lines 213-216 do not make sense. It appears the authors are contradicting their own findings. T2D is very well known to be inversely correlated with bone quality. The authors follow this statement with "nevertheless, DM was more common among participants with OP"....this should be as expected.

One of the major limitations to the study was pointed out at the end...that being that the type of exercise, be it aerobic versus resistance or weight bearing, are not reported in the analysis. It is, however, nice to see that exercise does represent a higher instance of non-OP participants than OP. 

Conclusion

Tables 1-5
This is a very large amount of data, which is informative to a point that it is overwhelming. To read and extrapolate on the data through pages of tabular results, however, is daunting. I would suggest that the authors consider placing some of this data in bar chart format, or something analogous to that. 

Were multiple DXA scans or questionnaires performed, as this might allude to the ability of exercise to maintain or improve bone, or reduce the impact of OP?

The authors should more accurately describe where "central body" scanning is for the DXA readings. 

Was any correlation run between obesity and BMD/T-scores?

Discussion
The statement in line 202 is very controversial. Increased body weight might correlate with increased bone, but many studies have indicated that increased weight as a consequence of obesity is not causal to increased bone mineral density, but detracts from bone quality. 

Lines 213-216 do not make sense. It appears the authors are contradicting their own findings. T2D is very well known to be inversely correlated with bone quality. The authors follow this statement with "nevertheless, DM was more common among participants with OP"....this should be as expected.

One of the major limitations to the study was pointed out at the end...that being that the type of exercise, be it aerobic versus resistance or weight bearing, are not reported in the analysis. It is, however, nice to see that exercise does represent a higher instance of non-OP participants than OP. 

Conclusion

Tables 1-5
This is a very large amount of data, which is informative to a point that it is overwhelming. To read and extrapolate on the data through pages of tabular results, however, is daunting. I would suggest that the authors consider placing some of this data in bar chart format, or something analogous to that. 
